# The Exergy Footprint as a Sustainability Indicator: An Application to the Neanderthal–Sapiens Competition in the Late Pleistocene

Enrico Sciubba

Department of Mechanical & Aerospace Engineering, University Roma Sapienza, 185 Rome, Italy;
enrico.sciubba@uniroma1.it

**Abstract:** A thermodynamic analysis of population dynamics and of sustainability provides rigor to many important issues. In this work, the "system society" is analysed in connection with the "system environment" using an exergy metric, and the method includes an internalization of the externalities (capital, labour, environmental effects) conducted on the basis of a "system + environment" balance. In this perspective, this study investigates the Late Pleistocene extinction of the *Homo neanderthalensis*, which took place in a geologically short time and in the presence of a competing species, the *Homo sapiens*. The case in study is not trivial, and its choice not casual: in those times, the only factor that could lead to an advantage of one group over the other was their respective resource use intensity. A specific indicator, the exergy footprint (EF), is here applied to measure the total amount of primary resources required to produce a certain (material or immaterial) commodity, including the resources needed for the physical survival of the individuals. On the basis of the available data, the results of a steady-state analysis show that the EF of the Neanderthal was higher than that of the Sapiens, and that with both species sharing the same ecological niche in a time of dwindling resources, the less frugal of the two was also more fragile in an evolutionary sense.

**Keywords:** extended exergy analysis; Neanderthal extinction; thermodynamic population models; sustainability; exergy footprint

---

## 1. Introduction: Thermodynamics, Sustainability, and Population Dynamics

The minimization of the adverse environmental effects of industrial activities, while maintaining the effectiveness of the "production chain", is a complex task that must rely on models that quantify the material and energy flows in a system [1]. Since a society is obviously composed of interacting individuals, it has been proposed to complement "industrial/ecological" models with a dynamic population model that accounts for the effects of the "production" on the numerosity of the population and vice versa. By doing so, it is possible to identify several types of interactions between the number of individuals, their needs, and the resources required to satisfy them. If such a combination of different approaches can explain with reasonable accuracy some of the large-scale events of the past, thus providing a sort of experimental validation to the models, and possibly reinforcing their mutual theoretical basis, its credibility in predicting the future may be reinforced [2].

The metaphor that human (industrial) societies "evolve" in a way similar to that of other natural ecosystems is gaining credibility and, given the growing throughput of anthropic structures, its application naturally rises concerns over the unsustainability of the current state of affairs. However, "sustainability" seems impervious to a rigorous and generally accepted definition: in 1996 a literature survey [3] identified almost 300 different *discursive* definitions of sustainability and sustainable development. It has been argued [4] that "the classical ecosystem analogy omits aspects of human

social and cultural life central to sustainability". According to the currently prevailing paradigm, "sustainable development" is thought to be resting on three pillars: economics, environment and society. In this (recently criticized) perspective, resource use is seen as a production factor inevitably increasing with numerosity, and the emphasis is shifted to its equitable, economical, and technologically feasible allocation.

Since drastic population limitation measures are out of the question, it is easy to see that the growing metabolism of the human society is approaching a critical limit, not only with respect to the available resource inputs, but—more importantly—to the sinks for waste and emission outflows. The so-called circular economy paradigm is a convincing strategy, which aims at reducing both the input of raw materials and the output of wastes by increasing as much as possible the amount of "recycling" (in its global sense, i.e., both of energy and of materials) [5]. Though the potential for "total recycling" is immense, very little attention is placed to the fact that recycling needs energy and (some) additional materials, and that Second Law of Thermodynamics places a limit to this "circularity" [6].

As for population dynamics, it is relatively easy today for anthropologists and paleo-archaeologists to identify direct links between geologically rapid changes in the numerosity of a population and the state of the environmental niche within which the population survives. While several studies successfully describe the history of the evolution or extinction of certain vegetal of animal species [7,8], it is extremely difficult to concoct a comprehensive model for the human society, where the interconnections among individuals, communities, and populations and between them and the environment are much more intricate.

In spite of the common idea that small (non-human) populations should be more affected by environmental change than large populations, experiments have shown that adaptability, maturity, genetics, and resource use appear to be as important factors as numerosity, and in fact small populations, at least at some life stage, may perform equally well or even better than large populations. It is only when they share the same ecological niche that relative numerosity may become an essential advantage [9]. If this is true for plants and lower vertebrates, it does not seem to apply to primates [10] (However, the idea that smaller human groups may over a sufficiently long time succumb to larger groups with whom they compete for resources ought to be tested by separating the "resource" from other more "anthropic" issues like technology, social organization, cultural habits ... is of course impossible for the contemporary human race). This led to the choice of the topic of this paper: since the competition between *Homo neanderthalensis* and *Homo sapiens* could not possibly be based on economics nor on environmental preservation issues, the only possible driver must have been the use they made of the available resources, which in a specific geological era, the late Pleistocene, were very scarce.

The existence of a quantifiable link between ecological dynamics and sustainability is of substantial importance here. On this point, some scholars argue that adaptive cycles are a fundamental property of living systems and that such systems can adapt to stresses in a manner such that each generation maintains properties experience proved to be healthy. Holling [11] defined sustainability as "the ability to create, test, and maintain adaptive capacity", and development as "the process of creating, testing, and maintaining opportunity". In his argument, he uses terms such as resilience, wealth, and opportunity to characterize an evolutionary path in which each generation retains many of the positive properties of the preceding one and possibly adds more desirable traits. He suggests that properly managed adaptive cycles constitute sustainable development, which is not at all the same "sustainable development" evoked by the familiar Brundtland definition. In spite of the merits of such an approach, I prefer using Ehrenfeld's definition: his framing of sustainability is that it is "the possibility that human and other forms of life will flourish on the planet forever. Flourishing has great metaphorical power" [12].

### 1.1. The Problem

The unusually fast disappearance of the *Homo neanderthalensis* has puzzled archeo-etnologists for a long time: What were the circumstances which drove to extinction in about 20,000 years a species that had survived for over 400,000 years, mostly in Europe, under extremely harsh environmental conditions, and in the course of time had evolved to an almost incredible degree of adaptation to such

conditions? Was it by chance that this extinction took place after Neanderthals and the primitive Sapiens started sharing their environmental niches? Was there a "war" between the two species, as implicitly hypothesized in several works [10,13]? Was there some sort of epidemic to which one of the species succumbed [14,15]? Did the large and frequent volcanic events of the late Pleistocene and the related strong and fast climate variations have an impact [16,17]? Some authors [18,19] blame the extinction of the Neanderthal species (HN) to the high rate of accidental mortality during their hunting; others [20,21], to their low fertility. This paper addresses the problem from a thermodynamic point of view, using well-published and accepted data to compile a reasonably accurate list of the primary resources available at that time and positing some assumptions—also extracted from reliable sources—on the final uses the two species made of the resources. Then, extended exergy accounting [22,23] is applied to two groups of the two species placed in the very same ecological niche (central Europe), to compare the "gross resource load" they placed on the environment. The striking conclusion emerges that, in each one of the five examined scenarios, the Neanderthal consumed a higher pro-capite amount of primary resources than the Sapiens, and this might have become a decisive factor when the two species came to coexist at a time of dwindling resources. It is likely that a dynamic study (i.e., one that includes in the calculations the variation in time of the climate and of the available resources) would provide some additional insight in this important issue: this topic is left for later studies. The steady-state study presented here confirms the importance of an exergy-based resource analysis method for the calculation of a new important thermodynamic sustainability indicator, the exergy footprint. A dynamic analysis that would include the effects of a variation in time of the relevant parameter (e.g., temperature and irradiation) and the birth rate/death rate balance, is of course much more complicated, and is not addressed here. Since similar studies do exist [19,23,24] and seem to confirm that a successful model can be constructed, it is suggested that the issue be tackled in future research.

### 1.2. The Exergy Footprint

The concept of the exergy footprint (EF in the following), introduced in [25], provides a rigorous thermodynamic basis to sustainability studies: it is a measure of the total primary resource consumption of a system (in particular, of a society), measured in terms of exergy. The use of exergy in lieu of energy has some advantages, both from a theoretical and from a practical point of view: some fundamentals are described in Section 3 below, and interested readers are referred to the vast literature on the topic (for example [21,26–28] and references therein). Anticipating the explanation provided in Section 3, it must be stressed that the fundamental difference between energy and exergy is that the former is conserved while the latter is not: in any physical process/transformation some exergy is destroyed by irreversibility. This confers to the primary exergy input the characteristic of a "cost" of the final exergy output. As a consequence, if one adopts the "final energy use" as a measure of resource consumption, the distinction between energy or matter streams of different quality (thermal vs. mechanical power, or chemical vs. thermal, etc.) is completely lost. What is of importance here is that societies with different life standards can be ranked according to their respective EF, a higher value indicating a "less sustainable" and a lower a "more sustainable" lifestyle. It is clear that in modern societies the interconnection of economic, social, political, and resource-related issues tends to blur the simple and rather blunt conclusion that a substantial change in societal structure of most advanced economies is necessary to reduce their EF: thus, a brave scholar who wished to propose a comparative EF study between two modern societies would find it extremely difficult to separate ethical, social and in general non-thermodynamic issues from the substantial result, i.e., from the inevitable exhortation to a more cautious resource exploitation. To simplify the issue, this paper presents a comparison between two pre-historic social organizations, for which sufficient data exist to compute the respective EF: demonstrating that the method works for primitive societies may lead to its successful application, mutatis mutandis, to contemporary ones. Since sustainability is obviously linked to political planning, it is believed that the new approach opens new possibilities for a rational comparison between "more" and "less" sustainable societal organizations.

### 1.3. Homo neanderthalensis and Homo sapiens

The origin of the Neanderthal species (HN in the following) is generally placed in the North-East part of Africa (roughly, today's Ethiopia and Kenya) around 500,000 years before present (yBP) [29]. Somewhat later, possibly 430,000 yBP, they migrated north, reaching Europe (some Neanderthal bones found in a cave in Spain in fact date back to 430,000 yBP [27,30]). They definitely came into contact with the *Homo heidelbergensis*, a more primitive species that appears to have preceded them in Europe about 600,000 yBP. The Neanderthal (HN) and the Sapiens (HS) migrations were not the first *Homo* migrations from Africa to Europe. Much earlier, different waves of *Homo* (*Homo erectus*, 1.9 million yBP; then *Homo heidelbergensis*, probably 600,000 yBP) had profited from prolonged periods of extreme monsoon activity in north-East Africa to cross the then-fertile and water-rich Sahara region [31,32]). Neanderthals adapted very well to the Pleistocene rigid climate and lived in small groups of hunters, but also knew some forms of gathering [29], knew the use of fire, and adopted a nomadic lifestyle. Their numerosity varied strongly with the successive glaciations, and in the relatively warm Saale/Weichel interglacial period (130,000–115,000 yBP (Figure 1) they went through a substantial demographic growth and spread into Eastern Europe. Approximately midway through the most severe of the four Pleistocene events, known as the Weichel glaciation (115,000–12,000 yBP, i.e., around 50,000 yBP), their numbers started to dwindle, and there are no known Neanderthalian settlements after 15,000 yBP.

The first traces of *Homo sapiens* (HS in the following) were found in the Ethiopian Omo Kibish region, a little North-East of the original site of appearance of the Neanderthal, and date approximately to 200–300,000 yBP [32,33]. An alternative opinion [34] is that the species evolved slowly but much earlier from the South-African predecessor *Homo rhodesiensis*. Be that as it may, it appears that they started migrating towards Europe only much later, about 110,000–70,000 yBP. There is also archaeological evidence of contact between the two species around 100–60,000 yBP, in what is now Syria, but in spite of some researchers' hypotheses, there is no certainty of interbreeding at that time [29,35,36]. It is certain though that the two species interbred at a later time: both archaeological findings [16,37] and DNA analysis [38] show that there was substantial interbreeding between HS males and HN females, while genetic analyses seem to prove that the HN male/HS female matings were not fertile [39–41]. Between 70 and 25,000 yBP the Sapiens colonized all Europe and most of the Russian subcontinent, reaching Asia and displacing local hominins populations, until 15–10,000 yBP when they crossed the Bering strait (at that time covered by a solid ice sheet), reaching into the Americas. The Sapiens were a hunter-gatherer society, practicing some primitive forms of agriculture [33,34] and herding, and were therefore less nomadic than Neanderthals. They lived in larger groups (over 30 members and up to 150 [33]) and were the first species to show "fast adaptive response", i.e., the capability of modifying their utensils (weapons, tools, pottery, etc.) in a relatively quick response to changed climatic conditions.

### 1.4. Neanderthal and Sapiens Lifestyles and Demographics

#### 1.4.1. Homo neanderthalensis

Fossil evidence shows that the extremely low population density (0.1 to 1 persons/km$^2$ [8,39,42]) and the relative abundance of prey strongly influenced the Neanderthal's technological and social characteristics in central and northern Europe. Small groups of HN ("family-based clans" rather than "tribes") displayed high residential mobility limited to rather short distances [29]. Their main prey were the herds of herbivores (horses, bison, reindeer, and snow goats), with a smaller amount of many other smaller game species. There is evidence that some Neanderthal groups in certain areas may have had a wider diet that included plants and smaller animals, but this pattern is not very widespread. In the southernmost coastal regions, their diet only occasionally included shellfish, birds, and turtles [43]. They scavenged and hunted, and their hunting was direct and dangerous: HN followed the herds, tried to isolate an individual prey and attacked it in groups by using wood-and-stone pikes and clubs (such hunting style explains the large amount of injuries evidenced by the fossil remnants). There appears to

have been no division of labour by gender between hunting and gathering, which means that both males and females (and probably younger adults) worked indifferently as hunters and gatherers. HN knew how to use fire both for cooking meat (and probably some vegetables) and for hardening the points of their pikes. Their nomadic or semi-nomadic production system, largely skewed towards hunting, was obviously affected by high incidental mortality [15,18], which is one of the factors that slowed the HN demographic growth rate. In addition, the metabolic expense associated with this lifestyle required a high-calorie diet (essentially game). Recent estimates [29,44] conclude that the metabolic cost of an adult Neanderthal was very high: 3500–5000 kcal/day (0.17–0.24 kW) vs. 2000–2400 kcal/d (0.09–0.12 kW) for a modern European human adult male. This, combined with the semi-nomadic lifestyle, probably caused long birth intervals in HN females and, therefore, low fertility. There are also indications that birth was difficult for HN females [40,41], and this has led to modern estimates of five or less children per female. It is also possible that, like the caribou females studied by Bårdsen [45], HN females confronted with more and more rigid winters adopted low-risk strategies by allocating more resources to building their own body reserves during summer and less to reproduction.

In the longest interglacial period, the warmer climate resulted in the extension of steppe-like savannah and grassland areas in eastern Europe, and in a large increase in the density of herbivores and of their carnivorous predators: this resulted in a demographic growth of HN, and led to their relatively rapid expansion towards the Black Sea and further east in the Russian tundra. It is also likely that this migration/expansion favoured a gradual and advantageous genetic homogenization, thanks to increased infra-breeding among different groups and the contemporary increase in population density. However the enlarged communication did not lead to quantum jumps in the lifestyle and, on the contrary, all experimental evidence demonstrates that HN were rather "technologically conservative", because their hunting techniques did not substantially change over more than 150000 years: the only consequence of the growth of the locally available prey was a reduction in their mobility. This in turn reduced the required caloric input and resulted in a rise in female fertility (more frequent pregnancies). There is no proof either of a change in their cultural lifestyle: they did not produce new tools, nor did they change their feeding habits. It is also unknown whether they still preferred to live in caves or natural shelters or had learned to build rough wooden huts like their predecessor, *Homo heidelbergensis*. As we shall see, this prolonged Neanderthal technological stagnation was one of the major causes of their low competitiveness with respect to the more technologically advanced Sapiens.

Recently proposed models [35,42,46,47] estimate an HN numerosity ranging from 5000 to 12,000 in their archaic periods (430,000 yBP), growing slowly to 75–100,000 individuals (1150,00 yBP), and then steadily declining from about 50,000 yBP to their extinction around 15,000 yBP.

These low numbers are better understood considering that except in few temperate niches (but semi-arid, except around the Mediterranean Sea), the vegetation was mainly cold steppe and tundra and was periodically devastated by an environmentally catastrophic alternation of warm (Dansgaard-Oeschger, DO) and cold (Heinrich, H) climate events. In paleoclimatic terms, the latter (H) is a period of very cold climate, while the former (D-O) is, on the contrary, a period of relatively warmer conditions. It is known today that such events, short in geological terms (less or about 1000 years), but very intense (variation of ±5–8 °C in the average yearly temperature), took place several times in the Pleistocene [20]. It can be thus reasonably assumed that the Neanderthal population, living in smaller groups under conditions of extreme and in geological terms rapid environmental instability, was primarily stagnant, with frequent genetic bottlenecks: random accidents (fires, floods, diseases, earthquakes, etc.) could lead to periodic episodes of decline, possibly accelerated by occasional skirmishes with their cousins the Sapiens [13].

### 1.4.2. Homo sapiens

It is quite apparent that, in contrast to the low rate of numerosity of the HN, there was a demographic advantage for *Homo sapiens*, at least during the window of observation of the present study. Such an advantage was probably the result of a higher fertility (due to a less nomadic lifestyle)

and/or reduced premature mortality (thanks to a safer hunting strategies), because no proof has been found that adult longevity or other structural causes have played a role in this demographic transition. Though there is no final proof of episodes of "tribal war", it is likely that, like other primates and hominins, whenever two groups happened to forage in the same environmental niche, some sort of direct physical confrontation may have arisen, leading to accidental deaths, mostly among youngsters and adult males. The median life span (25–40 years) and the maximum longevity (over 65) are proven to be almost the same between HN and HS [47].

The societal organization of the HS was quite different from that of the HN: they lived in larger groups (30–150 individuals [33]), had domesticated some animals (notably, the wolf), and were expert gatherers and hunted with spears, avoiding direct contact with the prey. There appears to have been a subdivision of tasks in the HS tribes, with "specialized" hunters, gatherers (probably, the females and youngsters) and toolmakers. This is in stark contrast with the "individualistic" HN members, who made their tools by themselves and divided their time between searching for suitable stone materials and wood for the weapons, gathering, hunting and preparing the meat. HS made and used stone tools, built mud-and-wood huts, invented a variety of conceptually complex and specialized items like composite stone tools, in later ages fishhooks and harpoons, bows and arrows, spear throwers and sewing needles. A strong hint to their lifestyle is the fact that already the first HS produced "consumers goods" like pottery (as early as 40,000 yBP), or stone and later ceramic figurines.

The HS diet was more varied than that of the HN, and their hunting technique less dangerous thanks to the use of projectile spears: this implies lower mortality of young adults and a higher demographic growth, favoured by the higher fertility of the HS females (more than five and up to ten children per female [39,47]). Early Sapiens lived apparently a more sedentary life than HN, and this probably favoured their cultural development, because their nomadism was likely to be seasonal rather than continuous.

## 2. Materials and Methods 1: The Late Pleistocene Climate

This section is necessary to better understand the development of the study, but it contains material clearly outside of my field of expertise. Therefore, I relied extensively on some excellent paleo-meteorological works, namely [14,17,19,20,32,48–51]. Possible inaccuracies are to be ascribed solely to my elementary understanding of climate processes.

### 2.1. The Glaciations

Paleo-climatological evidence shows that during the Pleistocene the temperatures in the Northern Hemisphere were substantially colder than today, while the now-tropical and sub-tropical areas enjoyed a humid and temperate climate [19,52]. The geographical region this study applies to is the portion of the European continent spanning from the Mediterranean coast to northern Scandinavia and the westernmost part of northern Russia. The first of two extended periods of extremely rigid climate was the Saale glaciation (named after the river Saale in northern Germany) that lasted from about 380,000 to 130,000 yBP. The HN survived this period, while the HS had not yet entered the scene. The second extended glaciation (Weichsel glaciation) lasted from about 115,000 to 12,000 yBP: this will contain—for reasons that shall be made clear shortly—our window of observation. Its name originates from that of a river in Poland where geological remnants were first discovered. In early Weichsel the glaciers were limited to large parts of Southern Scandinavia, while northern Europe was covered by tundra and low shrubs of birch and willow. The prevailing fauna included mammoth, woolly rhinoceros, bison, reindeer, musk ox, and their predators, but smaller herbivore and insectivore mammals were also present. This glaciation contained a brief but very cold period (23,000–19,000 yBP) known as the Last Glacial Maximum (LGM), in which the only area not covered by glaciers was that comprised between today's northern Europe and the Alps: this entire region was a large steppe devoid of natural barriers in which large ungulates and their predators were abundant [9]. Again, the coastal region of the Mediterranean Sea was somewhat warmer (average yearly temperature between −1 and 2 °C),

and covered by forests of pine trees. Experimental findings confirm that Europe was continuously inhabited already 600,000 yBP by the *Homo heidelbergensis*, a predecessor of the Neanderthal. When the latter migrated from northern Africa via the Middle East corridor, they colonized the Balcans, central, western and southern Europe and took advantage of some interglacial period to reach northern Russia (30–40,000 yBP) and probably pass the Ural Mountains as well.

## *2.2. Volcanism*

Pleistocene climate was affected by impressive volcanic activity that affected both the flora and fauna at various times. Such eruptions, denoted as Damskaard-Oeshger (D-O) events, left traces that are still easily identifiable in the sub-surface crust layers of Eurasia. In the most recent portion of our window of observation 75,000–15,000 yBP) the main event was the Toba eruption (Sumatra, 75,000 yBP), the largest ever recorded on Earth, which resulted in extremely high atmospheric dust concentrations over 10,000 years [14,20]. Another significant event was the almost as powerful Campanian Ignimbrite eruption (modern Napoli, Italy, 40,000 yBP) [17,48,49] that led to very high accumulated dust levels in the atmosphere, causing a century long "volcanic winter" and depositing millions of tons of dust over Italy and eastern Europe. These volcanic events contributed to a strong decrease in the average Earth temperature and to an equally large mortality of most of the then-existing plant species, which in turn caused a strong decrease in the animal population, especially the large herbivores.

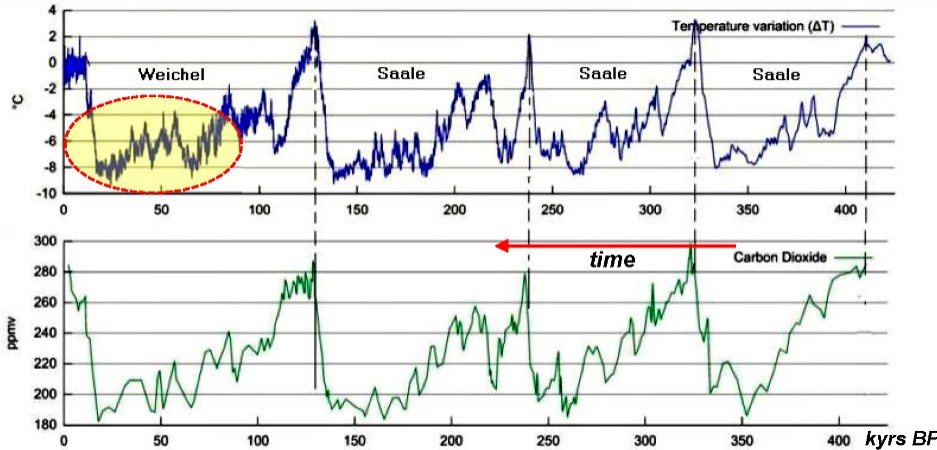

**Figure 1.** Variations in temperature with respect to the present (°C) and atmospheric $CO_2$ concentration (ppm) in the Pleistocene (adapted from [49]). Our window of observation is highlighted. Time progresses from right to left.

## 3. Materials and Methods 2: Exergy Analysis of the Neanderthal and Sapiens Societies

The main goal of this study is to quantify the (final and primary) resource use of the two species from a thermodynamic point of view. HN and HS were chosen because their respective societal organizations were rather elementary, and did not include any real "economics", which makes their societies a good benchmark for theories of exergetic cost. The idea is to calculate the amount of primary resource consumption needed by each group to produce their tools while following their characteristic lifestyle. The extended exergy accounting (EEA) method [25] was applied to "production systems" that model the Neanderthal and Sapiens societies, in an attempt to understand whether the composition of the resource basis and the consumption patterns of the two societies may give some hint as to the reason for the stunning and "sudden" (in geological times) prevalence of the one species over the other.

Exergy is defined as the maximum amount of work that can be ideally extracted from a system in an initial state A when it is allowed to freely relax to its thermal, mechanical and chemical equilibrium with the surrounding environment (in fact, other energy types, like magnetic, vibrational, electrical, nuclear . . . , are included in the general definition of exergy, and are neglected here because irrelevant

for the present analysis). It can be shown that a useful reversal of this definition also applies: exergy is the minimum amount of ideal work required to bring a system from its equilibrium state to any other state A. This second point of view convenient because it allows for the calculation of the "intrinsic" exergy content of substances found in nature: a stone composed of Ca, Mg, $H_2O$, S, and Si, for example, can be attributed an exergy content on the basis of this second definition [21]. In the context of the present study, each material existing in nature has its own exergy, and this is the "thermodynamic value" taken here for any input. Energy fluxes possess exergy as well: for solar irradiation the so-called Petela formula [21] is used, thermal energy Q has an exergy equal to $Q(1-T_0/T)$, etc. [28,53]. In extremely simplified terms, exergy analysis (ExA) includes the irreversible entropy production into its budget which is therefore is NOT conserved. Any real process is affected by an exergy destruction given by the equivalent of the Gouy–Stodola law. This is more rigorous than an energy or an entropy analysis in assessing the real thermodynamic "potential" that drives the relaxation to equilibrium [28,53].

An ExA consists in the identification and quantification of all fluxes entering and exiting a system: since exergy is not conserved (in each real processes a portion of it is destroyed by irreversibility), the ratio $P_j/F_k$, where the $P_j$ are the desired "products" and $F_k$ (the necessary "fuels") represents the exergetic efficiency of the process. EEA adopts—and extends to labour, capital, and environmental remediation costs—the symbolism suggested by Tsatsaronis [53]: the $P_j$ are the desired (or selected) final products of a process, and the $F_k$ all the inputs necessary to generate that set of $P_j$.

In 1999, an extension of ExA was proposed that consisted in the incorporation into the "exergy budget" ($E_{in}$, $E_{out}$) of the equivalent primary exergy required to support labour activities ($EE_L$), capital expenses ($EE_K$) and environmental remediation costs ($EE_O$): the method has seen promising applications to industrial processes and societal sectors [36]. For obvious reasons, this paradigm is known as "extended exergy analysis", or EEA. Previous studies on its application [22,26,54–56] have provided some interesting results: EEA leads to the calculation of the total amount of primary exergy needed to produce a material or immaterial commodity [22,26,57], i.e., to a *thermodynamic cost index*. More recently, it has been shown that the application of EEA to a society provides a measure of its "exergy footprint", i.e., of its global resource consumption, which is obviously an indicator of the thermodynamic sustainability of the "system society" (in more proper terms, an indicator of the degree of its unsustainability).

In conclusion, ExA has two advantages with respect to an energy or an entropy analysis: (1) it provides a uniform quantitative basis for the calculation of natural flows (whereas for instance chemical, thermal and mechanical energies are not directly comparable as to their "use value" to humans) and (2) it provides a direct quantification of the relative importance of irreversibilities (an entropy analysis reveals how large the irreversibility is in absolute terms, but does not provide per se a direct estimate of their relative importance i.e., of the ratio between the energy degradation rate $T_{ref}\Delta S$ and the energy flow through the process, $\Delta En$). The steady-state analysis is presented here (a dynamic analysis along the lines proposed in [23,24,58] requires a substantially larger database, and is left for future studies). It consists of two steps: for a given set of specifications that include the resource input, the numerosity of the two groups, their respective allocation of the workhours, and the output ("products"), the exergy flows through each system are calculated. An "exergy cost" is thus obtained, which has a pure thermodynamic value and provides at the same time a measure of the irreversibility in each production process and a verification of the credibility of the results. On this basis, labour intensity is factored in, and an extended exergy analysis is performed. Neither capital ($EE_K$) nor environmental remediation ($EE_O$) costs are accounted for in the calculations, given the absence of capital flows and the negligible influence of these primitive societies on the environment.

### 3.1. Resource Input

Although the numerical values are at best modest approximations of the real final resource use, we shall use the estimates given in Table 1 to derive the respective primary exergy consumption of the two groups. The following main model assumptions are posited:

(1) The meat intake was taken from [43], while the vegetable consumption (berries and roots) was assumed to be similar to those of some modern tundra nomadic tribes. Wood and stone consumption was calculated on the based on of the amount of tools produced and (for wood) to an approximate calculation of a 24 h/day campfire in a cave.

(2) The exergy content of live meat and vegetables is assumed equal to their respective average exergy content (conventionally measured by nutritionists in kcal/kg, here in kJ/kg), neglecting the intrinsic exergy value of the live prey and of the plant;

(3) The hunting, preparation, and transportation of meat, vegetables, stone, and wood are accounted for as equivalent primary labour exergy (i.e., they are not included in the exergy analysis but only in the EEA evaluation);

(4) The wood was gathered rather than cut, while the stone material is considered to be extracted from caves not necessarily in the vicinity of the HN or HS camps;

(5) The exergy of solar irradiation is a "hidden input" into the domestic sector, since it maintains the biosphere whence both species extracted their resources. Its value for the reference scenario is in accordance with accepted data [19,52,59].

### 3.2. The Neanderthal's and Sapien's Resource Basis

The resources exploited by the two species was almost the same: meat (a higher quantity pro capite for the NH), wood for cooking and heating, stones for weapons and utensils, and vegetables (a higher quantity pro capite for the HS). It is likely that both species camped in the immediate vicinity of water sources to facilitate supply. Well-accepted estimates for the food intake are available: the remaining values highlighted in grey in Table 1 are likely but not certain. The solar irradiation (400 W/m$^2$ for both groups) was calculated on an area of 5 km$^2$, assumed as the "reference living domain" for both species.

**Table 1.** Estimated final resource consumption of *Homo neanderthalensis* and *Homo sapiens*.

| Resource Type | Neanderthal (pro capite) | | Sapiens (pro capite) | |
|---|---|---|---|---|
| Meat | 3.44 kg/day | 20,608 kJ/day | 2.25 kg/day | 13,470 kJ/day |
| Wood | 5.34 kg/day | 26,700 kJ/day | 3.27 kg/day | 16,350 kJ/day |
| Stone | 0.86 kg/day | 12,900 kJ/day | 0.83 kg/day | 12,450 kJ/day |
| Vegetables | 0.10 kg/day | 340 kJ/day | 0.20 kg/day | 605 kJ/day |
| Other (mud, water) | 5 kg/day | 5000 kJ/day | 10 kg/day | 10,000 kJ/day |
| **TOTAL, kJ/day/person** | **65,548** | | **52,875** | |

Note: The kJ values in the third and fifth columns are the estimated exergy contents of the resources.

### 3.3. Internal Work Division

As stated above, the labour division was one of the major differences between the two species. The members of the groups are identified as "male adults" $N_M$, "female adults" $N_F$, "young adults" $N_Y$, "juveniles" $N_J$, and "old and injured" $N_{OI}$. The group numerosity is $N_{tot} = N_M + N_F + N_Y + N_J + N_{OI}$, and number of working members is $N_W = N_M + N_F + N_Y$ [60]. The value $N_W/N_{tot}$ is a model parameter whose effect on the result is substantial, though no convincing experimental data are available for it. The above numbers for both species are adapted from [39,42]. Table 2 reports the values assumed in this study. The $N_W$ assumes different values in the two societies, because while in the HN group all adults equally shared the work ($N_W = N_A$), in the HS tribe the labour tasks were differentiated and the number of injured hunters was percentagewise lower. In the following, for both groups it is assumed that females and youngsters took care of the cooking and gathering while the males prepared the weapons, and the relative percentages were extracted from [39]. As an additional assumption, the hunting was performed only by adults ($N_M + N_F$) [40]. The labour flows are calculated on the average working hours/day/person. An allowance was made for the fact that the better organization of the more numerous Sapiens society reduced their individual workload, so

that besides hunting, gathering, and toolmaking, the Sapiens also found time for "artistic" production: these hours are counted here as "idle" time.

**Table 2.** Estimated demographics for *Homo neanderthalensis* and *Homo sapiens* (number of individuals per clan or tribe).

| | $N_M$ | $N_F$ | $N_Y$ | $N_J$ | $N_{OI}$ | $N_{tot}$ | $N_W$ |
|---|---|---|---|---|---|---|---|
| Neanderthal | 2 | 4 | 3 | 2 | 2 | 13 | 9 |
| Sapiens | 10 | 20 | 15 | 10 | 5 | 60 | 45 |

### 3.4. Representative Products and Production Process

To perform an exergy analysis, it is necessary to model some production process. Considering the respective lifestyles described in Section 3, we shall compare the resource consumption required by the production of two indispensable artefacts common to the two societies: a stone axe (for both HN and HS) and a wood-and-stone pike (for HN) or spear (for HS), assuming different material and labour intensity for the two products. The respective production processes are depicted in Figure 2. To normalize the production with respect to the number of members of each group, the results are presented normalized with respect to $N_{tot}$ (i.e., in kJ per person per day). An allowance was made for the "leisure" activities of the HS (statuettes, ornaments, etc.) by reducing their workhour load by 1 h per day.

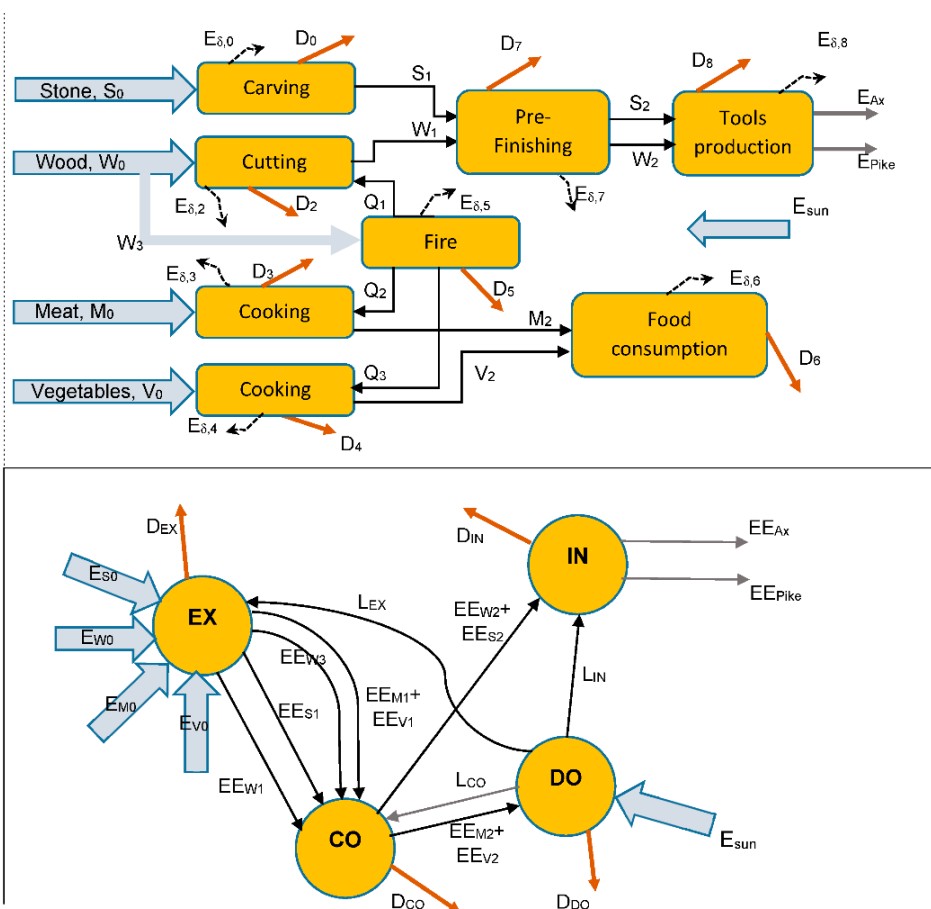

**Figure 2.** The physical exergy flows (top) and the extended exergy accounting (EEA) model of the Neanderthal species (HN) and HS production chains. Legend for Figure 2: D = discharge; $E_\delta$ = exergy destruction; M = meat; V = vegetables; Q = heat; $E_{sun}$ = exergy of solar radiation; L = Labour; S = stone; W = wood; CO = conversion sector; DO = domestic sector; EX = extraction sector; IN = industrial sector.

To draw any "balance", the boundary and the internal connectivity of the system must be defined. The upper part of Figure 2 represents the physical model of either society: stone and wood are inputs to the "carving" and "cutting" processes, and then go through a pre-finishing (gross sizing, fracturing, pruning etc.) and a final assembly line (axe and spear/pike construction). Meat and vegetables are processed (prepared and cooked) in separate "production lines". Some of the wood (W3) is not used for toolmaking but to cook and to provide "space heat". Cooked food (meat M2 and vegetables V2) is finally consumed by the tribe.

The bottom part of Figure 2 represents the extended exergy model of the same system. According to the EEA paradigm [25], clan and tribe activities are divided into four sectors: extraction, conversion, industrial, and domestic. In the EEA theory, there are three more sectors, clearly not applicable to the HS/HN societies: agricultural (AG), tertiary (TE), and transportation (TR). These somewhat overstated denominations refer to the collection of stone, wood, meat and vegetables (EX), to their preparation (CO), consumption (DO) and final use (IN), respectively. It is apparent that the physical model is at a much more disaggregated level than EEA: this is necessary though, because the exergy budget of the set of the individual sectors is described in terms of the individual fluxes of material and energy they exchange. EEA introduces three additional exergy fluxes ($L_{CO}$, $L_{EX}$, $L_{IN}$) that quantify the primary exergy equivalent of the workhours invested by the $N_W$ working members of the society in each production task. The system equations are derived in the following sections. Although the use of the terms "conversion" and "industrial" may appear inappropriate to describe the simple activities they refer to here, the names are the same as in the general EEA nomenclature, for ease of comparison with similar works.

### 3.5. Mass and Exergy Balances

With reference to the symbols used in Figure 2 top, the balance equations for each process are:

- *Carving*

$$
\begin{aligned}
\dot{m}_{S0} &= \frac{\dot{m}_{S1}}{(1-\psi_{0,S})} & \dot{E}_{S0} &= \dot{m}_{S0}e_{S0} \\
\dot{m}_{S1} &= \frac{\dot{m}_{S2}}{(1-\psi_{7,S})} & \dot{E}_{S1} &= \dot{m}_{S1}e_{S0} \\
& & \dot{E}_{S2} &= \dot{m}_{S2}e_{S0}
\end{aligned}
\tag{1}
$$

- *Cutting*

$$
\begin{aligned}
\dot{m}_{W0} &= \frac{(\dot{m}_{W1}+\dot{m}_{W3})}{(1-\psi_{0,W})} & \dot{E}_{W0} &= \dot{m}_{W0}e_{S0} \\
\dot{m}_{W2} &= \frac{(\dot{m}_{W1})}{(1-\psi_{7,W})} & \dot{E}_{W1} &= \dot{m}_{W1}e_{W0} \\
& & \dot{E}_{W2} &= \dot{m}_{W2}e_{W0}
\end{aligned}
\tag{2}
$$

- *Cooking*

$$
\begin{aligned}
& & & & \dot{E}_{M0} &= \dot{m}_{M0}e_{S0} \\
\dot{m}_{M0} &= \frac{\dot{m}_{M1}}{(1-\psi_{0,M})} & \dot{m}_{V0} &= \frac{\dot{m}_{V1}}{(1-\psi_{0,M})} & \dot{E}_{M1} &= \dot{m}_{M1}e_{S0} \\
\dot{m}_{M1} &= \frac{\dot{m}_{M2}}{(1-\psi_3)} & \dot{m}_{V1} &= \frac{\dot{m}_{V2}}{(1-\psi_4)} & \dot{E}_{M2} &= \dot{m}_{M2}e_{S0} \\
& & & & \dot{E}_{V1} &= \dot{m}_{V1}e_{V0} \\
& & & & \dot{E}_{V2} &= \dot{m}_{V2}e_{V0}
\end{aligned}
\tag{3}
$$

- *Feeding fire*

$$
\begin{aligned}
\dot{m}_{W3} &= \frac{(\dot{q}_1+\dot{q}_2+\dot{q}_3)}{[LHV_{W3}(1-\psi_5)]} & \dot{E}_{W3} &= \dot{m}_{W3}e_{W0} \\
& & \dot{E}_{qj} &= \dot{q}_j\eta_C
\end{aligned}
\tag{4}
$$

- *Feeding*

$$
\begin{aligned}
\dot{m}_{M2} &= \frac{\dot{m}_{M1}}{(1-\psi_{6,M})} & \dot{E}_{M2} &= \dot{m}_{M2}e_{M0} \\
\dot{m}_{V2} &= \frac{\dot{m}_{V1}}{(1-\psi_{6,V})} & \dot{E}_{V2} &= \dot{m}_{V2}e_{V0}
\end{aligned}
\tag{5}
$$

- *Production*

$$\dot{m}_{Ax} = \frac{(\dot{m}_{S2,ax}+\dot{m}_{W2ax})}{(1-\psi_{8,Ax})} \qquad \dot{m}_{pike} = \frac{(\dot{m}_{S2pike}+\dot{m}_{W2pike})}{(1-\psi_{8,pike})}$$
$$\dot{E}_{Ax} = \dot{m}_{S2Ax}e_{S0} + \dot{m}_{W2Ax}e_{W0} \qquad \dot{E}_{Pike} = \dot{m}_{S2Pike}e_{S0} + \dot{m}_{W2Pike}e_{W0} \tag{6}$$

The inputs $\dot{Ex}_{S2}, \dot{E}_{W2}, \dot{Ex}_{M2}, \dot{Ex}_{V2}$ can be back-calculated on the basis of the estimated weight of each weapon and of the metabolic rate of HN and HS [29,38], and thus the system (1)–(6) can be solved if the "material waste coefficients" $\psi_j$ are known: in this work, the latter have been assigned by adapting the values proposed by different sources (Table 3).

**Table 3.** EEA model assumptions for the Neanderthal and Sapiens societies.

| *Homo neanderthalensis* | | | | *Homo sapiens* | | | |
|---|---|---|---|---|---|---|---|
| $\phi_{fuel}$ | 0.4 | $\psi_4$ | 0.5 | $\phi_{fuel}$ | 0.4 | $\psi_4$ | 0.5 |
| $\psi_{0,S}$ | 0.3 | $\psi_5$ | 0.2 | $\psi_{0,S}$ | 0.4 | $\psi_5$ | 0.2 |
| $\psi_{0,W}$ | 0.2 | $\psi_{6,M}$ | 0.2 | $\psi_{0,W}$ | 0.2 | $\psi_{6,M}$ | 0.2 |
| $\psi_{0,M}$ | 0.25 | $\psi_2$ | 0.1 | $\psi_{0,M}$ | 0.25 | $\psi_2$ | 0.1 |
| $\psi_{0,V}$ | 0.2 | $\psi_{6,V}$ | 0.2 | $\psi_{0,V}$ | 0.2 | $\psi_{6,V}$ | 0.2 |
| $\psi_{1,S}$ | 0.2 | $\psi_{7,S}$ | 0.2 | $\psi_{1,S}$ | 0.2 | $\psi_{7,S}$ | 0.2 |
| $\psi_{1,W}$ | 0.1 | $\psi_{7,V}$ | 0.2 | $\psi_{1,W}$ | 0.2 | $\psi_{7,V}$ | 0.2 |
| $\psi_2$ | 0.2 | $\psi_{8,Ax}$ | 0.2 | $\psi_2$ | 0.2 | $\psi_{8,Ax}$ | 0.4 |

Legend: $\phi_{fuel}$ = (kg of dry manure and straw)/kg of wood; $\psi_j$ = waste coefficient, in $kg_{waste}/kg_{input}$.

The results provide the values of the primary gross resource consumption (mass and exergy values) for the Neanderthal and Sapiens society and are listed in Table 4 (the flows are identified in Figure 2). The efficiencies $\varepsilon_1, \varepsilon_2 \ldots \varepsilon_7$ result from the calculations.

**Table 4.** Primary resource consumption for the Neanderthal and Sapiens societies (*m* in kg/person/day; *E* in kJ/person/day).

| *Homo neanderthalensis* | | | | | |
|---|---|---|---|---|---|
| $m_{S0}$ | 1.92 | $E_{S0}$ | 28,760 | $\varepsilon_1$ | 0.7 |
| $m_{W0}$ | 7.62 | $E_{W0}$ | 38,109 | $\varepsilon_2$ | 0.8 |
| $m_{M0}$ | 6.12 | $E_{M0}$ | 36,636 | $\varepsilon_{(3+4)}$ | 0.795 |
| $m_{V0}$ | 0.16 | $E_{V0}$ | 425 | $\varepsilon_7$ | 0.84 |
| *Homo sapiens* | | | | | |
| $m_{S0}$ | 2.16 | $Ex_{S0}$ | 32,444 | $\varepsilon_1$ | 0.6 |
| $m_{W0}$ | 5.82 | $Ex_{W0}$ | 29,109 | $\varepsilon_2$ | 0.8 |
| $m_{M0}$ | 3.75 | $Ex_{M0}$ | 22,450 | $\varepsilon_{(3+4)}$ | 0.795 |
| $m_{V0}$ | 0.28 | $Ex_{B0}$ | 756 | $\varepsilon_7$ | 0.8 |

### 3.6. The EEA Balance Equations

There are yet additional terms that express the equivalent primary exergy of Labour and do not appear in the exergy budget. An additional set of equations, specific of EEA, must be considered. Referring the reader to [25,26,61] for a detailed description of the EEA paradigm, it suffices here to recall that the specific equivalent labour exergy is posited to be equal to a portion of the total incoming exergy flux:

$$\dot{ee}_L = \alpha \frac{\dot{E}_{tot,input}}{wh_{person}} \frac{N_{tot}}{N_W} \tag{7}$$

where $N_W$ is the number of workers and *wh* are the workhours per worker per unit time (here, per day): the rationale adopted in deriving Equation (7) is that the total resource input sustains the entire

population, and if the work is performed by a reduced number of members, their contribution is in effect used to sustain themselves and the non-working part of the group. The econometric coefficient $\alpha$ must be obtained experimentally, which in our case is obviously impossible: to avoid circular definitions, it was calculated here by including $\dot{e}e_L$ among the model unknowns, and then inverting Equations (14)–(17). Since extended exergy measures the "cost", expressed in equivalent primary resources, of the outputs of a system, it obeys a conservation equation. This is an important point: while exergy is a physical quantity and is not conserved, extended exergy represents the J of primary exergy embodied is each J of final exergy. It is therefore a cost to which a conservative formation equation applies.

$$\sum \dot{EE}_{products} = \sum \dot{EE}_{inputs} \tag{8}$$

By definition [25] the EE of generic stream $j$ (regardless whether an input or an output) is obtained by summing the contributions given by the material, energy and labour contributions:

$$\dot{EE}_j = \dot{EE}_{j,M} + \dot{EE}_{j,E} + \dot{EE}_{j,L} \qquad [\text{kJ}/\text{s}] \tag{9}$$

Equation (9) is applied to each of the four sectors EX, CO, IN, and DO. Under the assumption that the exergetic cost of the waste flows is equal to zero, the extended exergy (rate) budgets (kJ/s) can be written as:

- *Extraction sector EX*

$$\dot{EE}_{S1} + \dot{EE}_{W1} + \dot{EE}_{M1} + \dot{EE}_{V1} + \dot{EE}_{W3} = \dot{E}_{S0} + \dot{E}_{W0} + \dot{E}_{M0} + \dot{E}_{V0} + \dot{EE}_{L,EX} \tag{10}$$

- *Conversion sector CO*

$$\dot{EE}_{S2} + \dot{EE}_{W2} + \dot{EE}_{M2} + \dot{EE}_{V2} = \dot{EE}_{S1} + \dot{EE}_{W1} + \dot{EE}_{M1} + \dot{EE}_{V1} + \dot{EE}_{W3} + \dot{EE}_{L,CO} \tag{11}$$

- *Industrial sector IN*

$$\dot{EE}_{Ax} + \dot{EE}_{Pike} = \dot{EE}_{S2} + \dot{EE}_{W2} + \dot{EE}_{L,IN} \tag{12}$$

- *Domestic sector DO*

$$\dot{EE}_{L,EX} + \dot{EE}_{L,CO} + \dot{EE}_{L,IN} = \dot{E}_{sun} + \dot{EE}_{M2} + \dot{EE}_{V2} \tag{13}$$

The four EEA equations can be rewritten in terms of the $c_{ee,j}$, the extended exergy costs of each flow. The extended exergy cost $ee_{c,j}$ is defined as the cumulative exergy of primary resources consumed for the production of 1 kJ of exergy of product $j$. It is dimensionless (kJ/kJ).

$$\dot{E}_{S0} + \dot{E}_{W0} + \dot{E}_{M0} + \dot{E}_{V0} + ee_L \dot{N}_{hours,EX} = c_{ee,S1}\dot{E}_{S1} + c_{ee,W1}\dot{E}_{W1} + c_{ee,M1}\dot{E}_{M1} + c_{ee,V1}\dot{E}_{V1} + c_{ee,W3}\dot{E}_{W3} \tag{14}$$

$$c_{ee,S1}\dot{E}_{S1} + c_{ee,W1}\dot{E}_{W1} + c_{ee,M1}\dot{E}_{M1} + c_{ee,V1}\dot{E}_{V1} + c_{ee,W3}\dot{E}_{W3} + ee_L \dot{N}_{hours,CO} = c_{ee,S2}\dot{E}_{S2} + c_{ee,W2}\dot{E}_{W2} + c_{ee,M2}\dot{E}_{M2} + c_{ee,V2}\dot{E}_{V2} \tag{15}$$

$$c_{ee,S2}\dot{E}_{S2} + c_{ee,W2}\dot{E}_{W2} + ee_L \dot{N}_{hours,IN} = c_{ee,Ax}\dot{E}_{Ax} + c_{ee,Pike}\dot{E}_{Pike} \tag{16}$$

$$\dot{E}_{sun} + c_{ee,M2}\dot{E}_{M2} + c_{ee,V2}\dot{E}_{V2} = ee_L(\dot{N}_{hours,EX} + \dot{N}_{hours,CO} + \dot{N}_{hours,IN}) \tag{17}$$

To obtain the exergy footprint of the system it suffices to sum Equations (14)–(17) and obtain:

$$\dot{E}_{sun} + \dot{E}_{S0} + \dot{E}_{W0} + \dot{E}_{M0} + \dot{E}_{V0} = c_{ee,Ax}\dot{E}_{Ax} + c_{ee,Pike}\dot{E}_{Pike} \tag{18}$$

where the right hand side represents the total equivalent primary exergy entering the system, i.e. *the total of the biosphere resources consumed by the system for its survival*: by definition, its EF. Assuming equiallocation (i.e., $c_{ee,pike} = c_{ee,ax}$), the EFs for the two societies are reported in Table 5.

**Table 5.** Exergy footprint of each product and of the two societies.

| *Homo neanderthalensis* | | | *Homo sapiens* | | |
|---|---|---|---|---|---|
| *EF*, kJ/person/day | $EE_{ax}$, kJ/unit | $EE_{pike}$, kJ/unit | *EF*, kJ/person/day | $EE_{ax}$, kJ/unit | $EE_{spear}$, kJ/unit |
| 117,988 | 43,435 | 21,718 | 97,025 | 38,032 | 7226 |

An EEA analysis provides though much more insight than this: by solving the system (14–17) in the unknowns $c_{ee,j}$ it is possible to assess how each processing step affects the total cost.

Since there are 11 unknowns ($c_{ee,S1}$, $c_{ee,S2}$, $c_{ee,W1}$, $c_{ee,W2}$, $c_{ee,W3}$, $c_{ee,M1}$, $c_{ee,M2}$, $c_{ee,V1}$, $c_{ee,V2}$, $c_{ee,Ax}$, $c_{ee,Pike}$), we need to specify seven auxiliary equations to close the system. These equations must represent a physical relation between the extended exergy costs. Adapting to the case in study the allocation rules of thermo-economics [53], we posit:

$$c_{ee,W3} = 1 \quad \text{(unpreprocessed input)} \tag{19}$$

$$c_{ee,S1} = c_{ee,W1} \quad \text{(co − products)} \tag{20}$$

$$c_{ee,M1} = c_{ee,V1} \quad \text{(co − products)} \tag{21}$$

$$c_{ee,S1} + c_{ee,W1} + c_{ee,V1} = c_{ee,M1} \frac{hrs_{gathering}}{hrs_{hunting}} \quad \text{(Labour intensity prevails)} \tag{22}$$

$$c_{ee,S2} = c_{ee,W2} \quad \text{(co − products)} \tag{23}$$

$$c_{ee,M2} = c_{ee,V2} \quad \text{(co − products)} \tag{24}$$

$$c_{ee,Ax} = c_{ee,Pike} \quad \text{(co − products)} \tag{25}$$

and obtain the results shown in Table 6.

**Table 6.** Extended exergy cost $c_{ee}$ of the intermediate streams, $c_{ee}$ in kJ/kJ, $ee_L$ in kJ/workhour.

| *Homo neanderthalensis* | | | | *Homo sapiens* | | | |
|---|---|---|---|---|---|---|---|
| $c_{ee,S1}$ | $c_{ee,m1}$ | $c_{ee,Ax}$ | $ee_L$ | $c_{ee,S1}$ | $c_{ee,m1}$ | $c_{ee,AX}$ | $ee_L$ |
| 2.473 | 3.972 | 5.233 | 32787 | 2.274 | 3.724 | 4.659 | 25732 |

These results agree quite well with some recent values measured in modern underdeveloped societies [25,61].

## 4. Sensitivity Study: Five Additional Scenarios

The results displayed in Tables 5 and 6 indicate that the pro-capite resource consumption of the HN was higher than that of the HS. To gather additional insight, a sensitivity study was conducted by varying the irradiation (to simulate colder and warmer periods), the numerosity of the groups (to eliminate the possibility of a "size" influence on the results), the amount of meat input by the HN. Four additional scenarios were computed:

Scenario 1: All data as the reference case, but final meat intake by HN reduced to 2.5 kg/(person* day);
Scenario 2: All data as the reference case, but both groups have the same size (60 members);
Scenario 3: All data as the reference case, but solar irradiation increased to 450 W/m$^2$;
Scenario 4: All data as the reference case, but solar irradiation decreased to 350 W/m$^2$;
Scenario 5: "Combined effects" scenario: 60 members in each group, irradiation 450 W/m$^2$, meat intake by HN 2.5 kg/(person*day).

Table 7 and Figure 3 display the extended exergy costs $c_{ee}$ of one ax and one pike or spear: in accordance with the EEA accounting rules, the two are considered as co-products (Equation (25)), and

therefore are attributed the same cost. This does not mean that their final extended exergies, given by $EE_{ax} = c_{ee}*E_{ax}$; $EE_{pike} = c_{ee}*E_{pike}$ respectively, are the same.

**Table 7.** Extended exergy cost $c_{ee}$ [kJ/kJ] of one axe/pike (HN) and one axe/spear) HS).

|  | $c_{ee,ax} = c_{ee,pike}$, HN | $c_{ee,ax} = c_{ee,spear}$, HS | Variations with Respect to the Reference |
|---|---|---|---|
| Reference | 5.233 | 4.659 | / |
| scenario 1 | 4.536 | 4.659 | $M_{2,HN} = 2.5$ |
| scenario 2 | 5.736 | 4.659 | HN = 60 |
| scenario 3 | 5.311 | 4.679 | I = 450 |
| scenario 4 | 5.155 | 4.639 | I = 350 |
| scenario 5 | 3.965 | 4.679 | I = 450, HN = 60, $M_{2,HN} = 2.5$ |

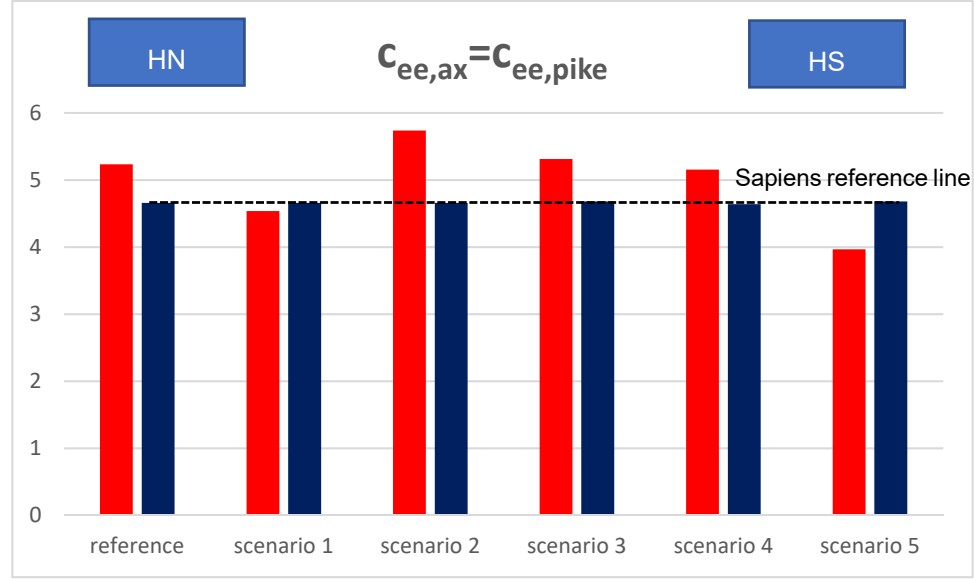

**Figure 3.** Extended exergy cost $ee_c$ of axe and pike under different scenarios.

Only under scenarios 2 and 5 the HN would have attained an evolutionary advantage in the resource consumption, with a very tiny one (2.6%) in scenario 1 and a quite robust one (15%) under scenario 5. But, of course, all paleoarchaeological evidence shows that HN never organized themselves in larger communities and never voluntarily modified their diet. A confirmation that the EEA model captures non-trivial correlations among different variables is provided by the non-additive effects of meat intake, numerosity, and irradiation on the result (scenario 5 vs. 1 + 2 + 3).

## 5. Do the above Results Support the "Unavoidable Extinction" Theory?

### 5.1. The Steady-State Perspective

The results (Table 6) show that the amount of primary resources needed to produce their weapons (one axe and one pike or spear) was different for the two species: *the average Neanderthal individual had to harvest more exergy from the environment to produce the same amount of final goods*. The reasons, thermodynamically speaking, are the above outlined different lifestyles and numerosity and the different allocation of labour inside of the group. The EEA model captures both the first difference (higher material resource consumption), and the resource intensity differential between the two production processes: notice that the environmental footprint of each artefact ($EF_{Ax}$, $EF_{pike}$, in kJ/unit) and that of the society as a whole ($EF_N$, $EF_S$, in kJ/(person*day)) are higher for the Neanderthals. Thus, the Neanderthals' individual primary resource consumption for the production of their hunting tools is substantially higher than that of their "cousins", the Sapiens. This is confirmed by the $EF_{ax}$ etc., that

quantifies the primary resources per unit. At a society level, the global amount of primary resources consumed by the HN to sustain themselves ($EF_N$, Table 5) was also substantially higher than that of the HS ($EF_S$).

The picture that emerges from these results provides the answer to the initial question: the Neanderthalian "technological production chain" was globally less efficient than the Sapiens' one; and furthermore, the Neanderthals' lifestyle placed higher requirements on the environment in terms of primary resources. It is then clear that, in case of dwindling resources (a situation that is very likely to have occurred in the Weichsel LGM and probably even before, during the post-Toba and post-Campanian Ignimbrite volcanic events), the HN species would have been at a strong disadvantage with respect to their more efficient competitors. Paleo-archaeological evidence shows in fact that in the period spanned by our window of observation the Neanderthals went through a series of "survival crises" that systematically decreased their numerosity and may have led to other fatal problems, like local genetic bottlenecks and low fertility [14,39,42,47], which sealed their extinction.

### 5.2. Limits of the Present Approach

As stated in the Introduction, the main intrinsic limitations of the model discussed in this paper are of two orders:

(1)　The scant available database, which leads to fundamental assumptions about the primary resource intake;

(2)　The steady-state assumption, that neglects important factors like birth/death rates, medium-range variations of the environmental data (irradiation, average ambient temperature, availability of prey).

In the current state of the art, the first problem cannot be remedied: it is only hoped that further paleo-ethnological research will shed more light on the actual resource input of the two species. More specifically, the final use estimates provided in Table 1 and the model assumptions about the "waste coefficients" shown in Table 3 strongly influence the results, and though the sensitivity study presented in Section 4 directly confirms the relative importance of some of the assumptions, there is no doubt that the validity of the results of an exergy footprint analysis essentially depends on the reliability of the database.

As for the approximations involved in the steady-state analysis, they are of course well known for any exergy-based (or not) analysis. First of all, it is obvious (as shown in Figure 1) that during the 50,000 years of our window of observation, both the average ambient temperature and the irradiation displayed short-term variations ($\pm 4\,°C$ and possibly $-150 \div 200$ W/m$^2$ due to volcanism), and this may have led to substantial changes in the local availability of prey (and in general of resources). Moreover, as discussed in [23], whenever two species coexist in the same environmental niche, the history of their numerosity is influenced not only by their individual resource use intensity, but also by other demographic factors (mainly, their respective birth/death balance).

The steady-state assumption was therefore an advisable first-approximation to check whether the model would work. Until a dynamic study is completed, the results reported here are to be taken as completely preliminary: they are surely rigorous (as the database allows) and indicate that a resource-based analysis is capable of producing reasonable results, but cannot be considered definitive.

## 6. Conclusions

Within the frame of a series of investigations aimed at the definition of thermodynamically-based sustainability criteria, the primary resource consumption of two primitive societies, the Neanderthal and the early Sapiens, have been compared by means of a steady-state model. The goal was to establish whether there were resource-related factors that placed the HN at a disadvantage with respect to their competitors. Under the given assumptions, the results show that in the considered window of observation between 115 and 15 kyBP the resource consumptions of the two species were indeed

substantially different, with the Sapiens' lifestyle being more "frugal". The inevitable simplifications required by the modelling and the strong dependence of the results on the initial assumptions suggest examining these results with care, although the sensitivity analysis presented in Section 4 seems to reinforce the idea that even significantly different scenarios would not have changed the Neanderthal's disadvantage. The analysis has been performed at steady-state, and in view of the likely influence of time-dependent factors like genetic bottlenecks, interbreeding, and possible local competition, this is a very strong limitation, as discussed in Section 5. It is therefore suggested that further attention to the dynamics of the exergy footprint of the two species be addressed in future studies.

In a broader perspective, the method applied here to the HN and HS societal competition may be, without substantial modifications, applied to modern societies, thanks to the availability of sufficiently disaggregate database for most of the contemporary countries. Though the derivation and the comparison of the exergy footprints of these much more intricately connected (and interconnected) systems may raise important issues in the correct allocation of "products" and "fuels" and require the utmost care in calculating the labour and capital equivalent primary exergy ($EE_L$ and $EE_K$), a complete standardization of the method is absolutely feasible. As remarked in the Introduction, the calculation of a general (un)sustainability metric is of the utmost importance to reach a consensus about our "path to a less unsustainable future".

**Conflicts of Interest:** The author declares no conflict of interest.

## Nomenclature

| Symbol: Units | Definition | Symbol: Units | Definition |
| --- | --- | --- | --- |
| $c_{ee}$ | Extended exergy cost, J/J | HN | Homo Neanderthalensis |
| CO | Conversion sector | IN | Production sector |
| D, kg/day | Discharge (waste) flux | L, work h/day | Work flux |
| D-O | Damskaard-Oeshger event | LGM | Last glacial maximum |
| DO | Domestic sector | $\dot{m}$, kg/day | Mass flow rate |
| E, J; e, J/kg | Exergy/specific exergy | M, kg/day | Meat flux |
| $\dot{E}_\delta$, W | Exergy destruction rate | N | Number of individuals |
| $\dot{E}$, W | Exergy flow rate | P, units/day | Product flux |
| EE, W; | Extended exergy | Q, W | Thermal energy flow |
| $ee_L$, J/workhr | EE of Labour | S, kg/day | Stone material flux |
| EEA | Extended Exergy Accounting | T, K | temperature |
| $E_{sun}$, W/m$^2$ | Net exergy of solar irradiation | V, kg/day | Vegetables flux |
| EX | Extraction sector | W, kg/day | Wood flux |
| F, kg/day | Food flux | yBP, years | Years before present |
| H | Heinrich event | | |
| **Greek Symbols** | | **Suffixes** | |
| | | ref | Environment state |
| | | S1 … , D1 … W1 … | Stream S1, D1, W1 … |
| δ | Destruction | | |
| ε | Exergy efficiency | | |
| ϕ | Fuel mixture ratio | | |
| ψ | Waste coefficient | | |

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
