# Peer review of "The Exergy Footprint as a Sustainability Indicator: An Application to the Neanderthal–Sapiens Competition in the Late Pleistocene"

_sustainability, doi:10.3390/su11184913_

Round 1

Reviewer 1 Report

This is a very interesting and intellectually stimulating paper. Since the subject matter is thousands of years in the past, there will of course be those who will disagree with the data used, the assumptions made, the methodology applied, and consequently the conclusions reached. Such criticism is to be expected, but should not be reason to dismiss or detract from the value of the paper.

To make the paper more robust, I recommend to add some more discussion about the other theories regarding the extinction of Neanderthal, and the role the Sapiens in this. There is an excellent discussion in Wikipedia. In the light of that, the statement in the abstract "in those times, the only factor that could lead to an advantage of one group over the other was their respective resource use intensity", and other similar statements (e.g. lines 74-76) need to be modified and possibly replaced by a more comprehensive discussion.

There are few other places that need attention:

- In footnote 4 "substantial interbreeding between HS males and HS females" should probably be "substantial interbreeding between HS males and HN females"

- The data in Table 1 does not justify the statement in lines 282-283. The difference in vegetable intake doesn't come close to making up for the difference in the meat intake. Also, the difference in the total is a substantial 25%.

- It would be useful to provide some explanation/justification for the estimates given in Table 1.

- A final review of the text would be useful to remove small typo and usage errors.

Author Response

please see rebuttal file

Reviewer 2 Report

This paper describes an intriguing topic, for different reasons.  It is original, and it presents a nice case of combining things which are not often seen in combination (archaeology and exergy). It is also intriguing that many papers in Sustainability are forward-looking, while this paper is backward-looking, in the far distant past.

At the same time, the paper shows several weak points. The major weak point is the sudden break in 4.4. Before 4.4, the paper summarizes the state-of-the-art in archaeological results on Sapiens and Neandertalensis. I am not an expert in this field, so I can't judge the correctness of what is written, but I did enjoy reading it. In 4.4, there is a sudden transition to a mathematical model. I am an expert in mathematics, so while this part will frighten some readers it actually attracted my attention. However, it is quite unreadable. The section hardly contains sentences, it just starts with "carving" and then a formula, the reader has little idea of what is happening). Worse, the symbols are unexplained, not in 4.4, but also not in the Nomenclature table on page 1. What is m dot? what is psi? What is E dot? What do the subscripts "S0", "S1", "0,S" etc refer to? How does figure 2 relate to such a formula? Of course, this should be added to the text. But interpretation is equally important. Most readers of Sustainability know little or nothing of exergy (I do). Now, the reader receives a layman explanation in lines 91-110, but the details (4.4) are obscure, to say it mildly. In fact, the confusion starts already a bit before 4.4, where I read about the "econometric coefficients alpha and beta" (line 304) and "eeL" (line 305)

Another comment is that the structure of the article is a bit chaotic. I would expect a research question around line 90, but it is now hidden in the midst of a paragraph (lines 119-125). A second part of the reserach question is in line 293 and a third in line 300. Another thing is that there are three sections Materials and Methods, but I think this is quitre misplaced. What is the Materials and Methods aspect of section 2? And what is section 2 different in character from section 1.2? I would suggest a split into sections titled Introduction (formulating the research question), Model (mainly exergy footprint), Data (for exergy, but also for demographics), Results, Conclusion.

Language is generally OK, although there are a few mistakes (e.g., line 30 "by"->"of", footnote in "discursiive", line 113 "of" is missing, line 147 "than"->"of", line 200 "lyfestyle").

There are quite some reference, but there are also many places which beg for a reference. Examples: recently criticized (lines 45-46), "common idea" (line 64), "experiments have shown" (line 65), "appear to be" (line 66), "well-accepted" (line 285), the numbers in lines 286-288 and Table 1, Table 2.

The text often speaks of "experiments" (e.g., line 181, 234). As far as I know, no experiments on Neandertals have been carried out. I think this must be "empirical".

The text also treats Sapiens as a species from the past. Sapiens "was", "did", "lived". Moreover they have a prehistoric "rather elementary" (line 295) life: domesticating the wolf, gathering and hunting. But I am a Sapiens as well. Please demarcate that you are speaking about a past-sapiens.

Reviewer 3 Report

This manuscript presents an assessment of the resource use intensity of Homo Neandertalensis and Homo Sapiens in the late Pleistocene, using exergy metrics. The results show that a higher resource requirements of the Homo Neandertalensis to sustain their lifestyles were a potential source of disadvantage in their competition with the Homo Sapiens for the same ecological niche.

It was a pleasure to read this manuscript, which is very clearly written, and provides an original perspective of the importance of resource efficiency metrics. I have, however, a few suggestions to clarify the presentation of the results and its contribution for sustainability assessments. Please find them below:

1.             In the first 5 paragraphs of section 1, the author provides a compelling discussion about the need for the assessment presented in this manuscript. However, it is unclear whether there are already any similar exergy assessments comparing Homo Neandertalensis with Homo Sapiens. Claiming the existence of such a gap in existing literature would strengthen the novelty claims of this manuscript.

2.             In lines 94–99 the author points to other references when justifying the use of exergy instead of energy metrics for this assessment. However, a brief clarification of the main arguments used to justify that choice should be included here. I agree with the author that the ranking of a resource use footprint is important for the context of this work. However, any other energy or resource footprint indicator would potentially serve the same purposes. Therefore, why is exergy the most appropriate metric for this particular assessment? Is it because it enables the quantification of energy and material flows in the same unit? If so, this should be clarified.

3.             A very good discussion on the potential important of the analysis offered in this manuscript to the field of sustainability assessment is provided in the introduction, but this discussion is absent from the conclusions. I would expect to see a discussion about the insights that the particular results and methods presented in this paper may provide to the assessment of current human sustainability challenges.

Please find below a list of minor comments:

4.             The results in Tables 1, 5 and 6 are presented with up to 6 significant figures. Given the obvious uncertainties associated with this estimation exercise, I would recommend the use of fewer significant figures in the results.

5.             What are the units of the demographics presented in Table 2? Thousands? Millions? The unit should be explicitly stated in the table or in its caption.

6.             In equations 9 to 25, many different variables and indexes are used. I suggest reminding the readers to report to the initial nomenclature table for these definitions. It would be helpful also to repeat these in the caption of Tables 5 and 6.

7.             Line 469: delete the first “the”.

8.             Line 496: perhaps “results” instead of “result”.

9.             The assessment presented in this paper conveys obvious uncertainties, and the author claims that a sensitivity analysis not presented here shows that the results stand despite the uncertainties in the assumptions. It would be very helpful if this sensitivity analysis could be presented here, even as a supplementary file, to strengthen the robustness of the results.

Round 2

Reviewer 2 Report

I am a bit disappointed in the careless way the author has revised his manuscript. It looks as if he has sent a draft, not a final version. Some examples are:
* line 314: "[szargut]" instead of "[53]" (or whatever number this is supposed to be)
* line 318: "[x,xx]"
* a very weird use of capitalized words, like in line 6 "A Thermodynamic analysis of Pupulation Dynamics and of Sustainability ..." or line 10 "Authors" or line 601 "Countries". On the other hand, Exergy Footprint in line 1 capitalized and in line 22 only partly.
* a nomenclature table which contains uniformative entries, like that alpha means alpha and beta means beta
* an inconsistent and confusing nomenclature. For example, E is exergy in W (are you sureit is not joule?), E_delta is exergy destruction in W (why no dot on E?), E^dot is exergy flow rate in kW (why this one in kilowatt instead of watt?), EE is extended energy in W (energy in watt?), etc.
* section 2 with the title "materials and methods 1" contains no materials and no methods
* section 3 has the title "materials and methods 3", which should have been renumbered to "2"
* line 516: why "hrs_gathering" in plural and "hr_hunting" in singular?
* there are two sections "4": line 529 and line 558
* line 607 gives a quote ("path to a less unsustainable future") which is supposed to be "remarked in the Introduction", but it is not.
* the reference list contains quite a few sloppy things. Some examples: [6] "PNAS" abbreviated, [7] author names underlined, [21] volume and page numbers missing, [22] "Frosh" is misspelled and "1989" is not the correct year
I kindly ask the author to carefully cross-check the document. Only then it is time for the reviewers.

Author Response

I have incorporated the suggested corrections in the second revision. Will submit it together with a full rebuttal (including the remarks of other reviewers)

Reviewer 3 Report

The author has completed a thorough review of the manuscript and addressed all my concerns. Congratulations for a very good paper.

Very minor comment: closing brackets missing in line 593.

Author Response

Thanks. A second revision will be uploaded together with a complete rebuttal (to all Reviewers' remarks)